

# Hybrid post-quantum Transport Layer Security formal analysis in Maude-NPA and its parallel version

Duong Dinh Tran[1], Canh Minh Do[1], Santiago Escobar[2] and Kazuhiro Ogata[1]

[1] Japan Advanced Institute of Science and Technology, Nomi, Japan
[2] Universidad Politécnica de Valencia, Valencia, Spain

## ABSTRACT

This article presents a security formal analysis of the hybrid post-quantum Transport Layer Security (TLS) protocol, a quantum-resistant version of the TLS protocol proposed by Amazon Web Services as a precaution in dealing with future attacks from quantum computers. In addition to a classical key exchange algorithm, the proposed protocol uses a post-quantum key encapsulation mechanism, which is believed invulnerable under quantum computers, so the protocol's key negotiation is called the hybrid key exchange scheme. One of our assumptions about the intruder's capabilities is that the intruder is able to break the security of the classical key exchange algorithm by utilizing the power of large quantum computers. For the formal analysis, we use Maude-NPA and a parallel version of Maude-NPA (called Par-Maude-NPA) to conduct experiments. The security properties under analysis are (1) the secrecy property of the shared secret key established between two honest principals with the classical key exchange algorithm, (2) a similar secrecy property but with the post-quantum key encapsulation mechanism, and (3) the authentication property. Given the time limit T = 1,722 h (72 days), Par-Maude-NPA found a counterexample of (1) at depth 12 in T, while Maude-NPA did not find it in T. At the same time T, Par-Maude-NPA did not find any counterexamples of (2) and (3) up to depths 12 and 18, respectively, and neither did Maude-NPA. Therefore, the protocol does not enjoy (1), while it enjoys (2) and (3) up to depths 12 and 18, respectively. Subsequently, the secrecy property of the master secret holds for the protocol up to depth 12.

# INTRODUCTION

As an early precaution in dealing with future attacks from quantum computers, extensive research efforts have been spent to construct post-quantum cryptographic protocols, replacing classical cryptographic protocols. This has been motivated by the fact that most of the asymmetric cryptosystems used today will be no longer secure under large-scale quantum computers. The reason is that the computationally hard mathematical problems on which these cryptosystems are relying can be efficiently solved by a sufficiently large quantum computer even though they are intractable for conventional computers. For instance, the integer factorization problem is no longer hard under a large-scale quantum

Corresponding author
Duong Dinh Tran,
duongtd@jaist.ac.jp

computer running Shor's algorithm (*Shor, 1994*). In recent years, with the involvement of many giants, such as Intel, IBM, and Google, different and stronger quantum computers have been introduced, making a practical one closer to reality. That motivates many cryptography and security research groups to investigate on building new post-quantum cryptographic protocols as well as analyzing their security.

In 2019, the Amazon Web Services (AWS) team proposed a quantum-resistant version of the Transport Layer Security (TLS) 1.2 protocol (*Rescorla & Dierks, 2008*), namely the hybrid post-quantum TLS Protocol (*Campagna & Crockett, 2021*), where TLS is known as one of the most important cryptographic protocols, protecting numerous communications over the Internet every day. Two parallel key exchange schemes are employed in the proposed protocol, namely a classical key exchange algorithm and a post-quantum key encapsulation mechanism (KEM), which is the reason why the word "hybrid" is used in its name. In the proposal, the classical key exchange algorithm is fixed as Elliptic Curve Diffie-Hellman (ECDH), while the post-quantum KEM (PQ KEM) can be, for example, Kyber (*Bos et al., 2018*) (precisely CRYSTALS-Kyber) and BIKE (*Aragon et al., 2019*). That hybrid scheme is expected to make a negotiated secret key at least as secure as ECDH against a classical attacker and at least as secure as the selected PQ KEM against a quantum attacker.

Our previous work (*Tran et al., 2022*) presented the Maude-NPA formal specification of the hybrid post-quantum TLS protocol, where Maude-NPA (*Escobar, Meadows & Meseguer, 2006*) is a powerful tool for analyzing cryptographic protocols. Maude-NPA is implemented in Maude (*Durán et al., 2020*), a specification/programming language based on rewriting logic, and supports an unbounded number of session executions as well as protocol participants. For modeling intruders' capabilities, the tool uses the Dolev-Yao intruder model (*Dolev & Yao, 1983*) and the strand model (*Thayer, Herzog & Guttman, 1998*). In this manner, the intruder is given the capability of fully controlling the network, for example, intercepting & modifying messages and impersonating some protocol participants to send some messages to other participants. Maude-NPA is based on narrowing and backward search. Narrowing is a generalization of term rewriting that allows variables in subject terms and replaces pattern matching by unification. The backward search starts from a given insecure pattern, *i.e.*, an attack pattern, representing insecure states, and runs backwardly to check whether it is reachable from an initial state. If an initial state is found, the attack concerned is a valid attack on the protocol; otherwise, the protocol is secure against the attack. Several optimization techniques to reduce the search space have been developed (see the article by *Escobar, Meadows & Meseguer (2008)*).

This article presents a formal analysis of the hybrid post-quantum TLS protocol with Maude-NPA. This is an extended version of our previous work (*Tran et al., 2022*), which provided a first attempt at formally specifying the protocol in Maude-NPA. The analysis belongs to the symbolic approach, which is typically preferred by formal method researchers. Whereas, its complementary, namely the computational approach, is often used by cryptographers. A computational proof gives a tighter security guarantee because it takes probability and complexity into account, but it is not easy to understand for non-experts in cryptography, and hard to mechanize the proof. A symbolic verification

is easier to understand, computer-verified, and suitable for automation, but it often needs to make some idealizing assumptions. We conduct the formal analysis with not only Maude-NPA but also a parallel version of Maude-NPA (called Par-Maude-NPA for short) (*Do et al., 2022*) to make the best use of multicore architectures.

In summary, our contributions in this work include the Maude-NPA formal specification of the Hybrid PQ TLS and the formal analysis of three security properties:

(1) The secrecy property of the ECDH shared secret key established between two honest principals. Par-Maude-NPA found a counterexample of this property at depth 12 in the time T = 1722 h (72 days), while Maude-NPA did not find it in T.

(2) The secrecy property of the PQ KEM shared secret key established between two honest principals. As the same time T, Par-Maude-NPA did not find any counterexample of this property up to depth 12, and neither did Maude-NPA up to depth 10.

(3) The authentication property, which states that if client *A* has completed the handshake, apparently with server *B*, then *B* is the actual principal who has communicated with *A*. As the same time T, Par-Maude-NPA did not find any counterexample up to depth 18, and neither did Maude-NPA up to depth 17.

The analysis results confirm that the protocol enjoys (2) and (3) up to depths 12 and 18, respectively, while it does not enjoy (1). Subsequently, the secrecy property of the master secret holds for the protocol up to depth 12. By using Par-Maude-NPA, we achieve a better running performance than the original tool in terms of analysis time. Hybrid PQ TLS is one large case study formalized in Maude-NPA. Through this work, we also would like to show a case in which Par-Maude-NPA is capable of detecting an attack, while Maude-NPA fails to do so due to an excessively long analysis time, which has not been demonstrated in *Do et al. (2022)*. The complete protocol specification and the attack patterns reported in this article can be found at https://doi.org/10.5281/zenodo.7919153.

In the remainings, we give some preliminaries related to Maude-NPA and Maude in Section 'Preliminaries'. We describe the exchange messages in the Hybrid PQ TLS protocol in Section 'Hybrid Post-Quantum TLS 1.2' and the Maude-NPA formal specification of the protocol in Section 'Protocol formal specification in Maude-NPA'. Afterward, Section 'Analysis experiments' reports the results of the experiments we have conducted with Maude-NPA and Par-Maude-NPA. Finally, Section 'Related work' mentions some related case studies and Section 'Conclusion' summarizes our article.

## PRELIMINARIES

Maude-NPA (*Escobar, Meadows & Meseguer, 2006*) is implemented in Maude (*Durán et al., 2020*), a declarative language and high-performance rewriting logic tool for specifying, programming, and verifying programs/systems, especially concurrent programs/systems. Maude supports both order-sorted equational logic and rewriting logic (*Meseguer, 2010*). Several formal analysis facilities, such as reachability analysis and LTL model checking, are implemented in Maude, making it possible to specify and analyze complex systems. This section briefly introduces the syntax of the Maude language (see the article by *Durán et al. (2020)* and the book by *Clavel et al. (2007)* for more details), how narrowing works, and how Par-Maude-NPA was parallelized.

### Functional modules

A functional module $\mathcal{M}$ specifies an order-sorted equational logic theory $(\Sigma, E)$ with the syntax: `fmod` $\mathcal{M}$ `is` $(\Sigma, E)$ `endfm` . $\Sigma$ is an order-sorted signature, which may contain a set of the following declarations:

- importations of previously defined modules (`protecting` ... or `extending` ... or `including` ...)
- sorts and subsorts (`sort` $s$ . or `sorts` $s\,s'$ . or `subsort` $s < s'$ . )
- function symbols (`op` $f$ : $s_1 \ldots s_n \rightarrow s\,[att_1\,\ldots\,att_k]$ . )
- variables (`vars` $v\,v'$ . )

where $s$ and $s'$ are sort names; $v$ and $v'$ are variable names; and $att_1, \ldots, att_k$ are equational attributes, such as `assoc` for associativity and `comm` for commutativity.

$E$ may contain a collection of equations, which may be either unconditional (`eq` $t = t'$ . ) or conditional (`ceq` $t = t'$ `if` $cond$ . ), where $t$ and $t'$ are terms and $cond$ is a conjunction of equations (*e.g.*, $t = t'$). Equations are used as *equational rules* to perform the simplification in which instances of the left-hand side pattern that match subterms of a subject term are replaced by the corresponding instances of the right-hand side. The process is called *term rewriting* and the result of simplifying a term is called its *normal form*.

### System modules

A system module $\mathcal{R}$ specifies a rewrite theory $(\Sigma, E, R)$ with the syntax: `mod` $\mathcal{R}$ `is` $(\Sigma, E, R)$ `endm` . $\Sigma$ and $E$ are the same as those in an equational theory. $R$ may contain a collection of rewrite rules, which may be either unconditional (`rl` [ $label$ ] : $u => v$ . ) or conditional (`crl` [ $label$ ] : $u => v$ `if` $cond$ . ), where $label$ is a name; $u$ and $v$ are terms; and $cond$ is a conjunction of equations and/or rewrites (*e.g.*, $t => t'$). Rewrite rules are also computed by rewriting from left to right modulo the equations in the system module and regarded as *local transition rules*, making many possible state transitions from a given state in a concurrent system.

### Narrowing

Narrowing is a generalization of term rewriting that allows logical variables in terms and replaces pattern matching by unification. To describe how narrowing works, in the following, we use a classical example in the Maude book (*Clavel et al., 2007*). The following system module specifies a concurrent machine to buy cakes (`c` ) and apples (`a` ) with dollars ($) and quarters (`q` ). A cake costs a dollar while an apple costs three quarters (specified by the rewrite rules `buy-c` and `buy-a` , respectively). Only dollars are allowed to buy cakes and apples. However, the machine can change four quarters into a dollar (the equation `change`). The complete specification of the machine is as follows:

```
mod NARROWING-VENDING-MACHINE is
  sorts Coin Item Marking Money State .
  subsort Coin < Money .
  op empty : -> Money .
  op __ : Money Money -> Money [assoc comm id: empty] .
  subsort Money Item < Marking .
  op __ : Marking Marking -> Marking [assoc comm id: empty] .
  op <_> : Marking -> State .
  ops $ q : -> Coin .
  ops c a : -> Item .
  var M : Marking .
  rl [buy-a] : < M $ > => < M a q > [narrowing] .
  rl [buy-c] : < M $ > => < M c > [narrowing] .
  eq [change] : q q q q M = $ M [variant] .
endm
```

where the operator `<_>` specifies the machine state. The attributes in the `__` operators
state that they are associative and commutative and have the identity element `empty` .
The attributes `narrowing` and `variant` are specially used for narrowing and variant-based
equational unification algorithms (*Escobar, Sasse & Meseguer, 2010*), *i.e.*, only rewrite rules
and equations with these attributes are used to perform the algorithms. Let us consider a
term `< M1 >` as an initial state that only contains a variable of the sort `Money` . There would
be several traces from the initial state by using narrowing. At each narrowing step, we must
choose which subterm of the subject term, which rewrite rule of the specification, and
which instantiation on the variables of the subterm and the left-hand side of the rewrite
rule (or which unifier (or substitution) of the subterm and the left-hand side of the rewrite
rule) are going to be considered. Each narrowing step applied to a given state produces a
new branch in the reachability tree. For example, for each rewrite rule of the machine, there
is only one unifier that makes the initial state `< M1 >` equal the left-hand side of the rewrite
rule. Therefore, with one narrowing step, there are only two possible cases, producing two
successor states from the initial state as follows:

$$< \text{M1} > \leadsto_{\sigma_1, \text{ buy-a}} < \text{a q M2} >$$
$$< \text{M1} > \leadsto_{\sigma_1', \text{ buy-c}} < \text{c M2'} >$$

where `M2` and `M2'` are variables of the sort `Money` and the substitutions are $\sigma_1 = $ `M1` $\mapsto$ `M2`, `M`
$\mapsto$ `M2` and $\sigma_1' = $ `M1` $\mapsto$ `M2'` , `M` $\mapsto$ `M2'` with the rewrite rules `buy-a` and `buy-c` , respectively.
Note that `M` in the substitutions is the variable used in the left-hand side of the rewrite
rules. From the successor state `< a q M2 >` , two more consecutive narrowing steps may be
performed as follows:

$$< \text{M1} > \leadsto_{\sigma_1, \text{ buy-a}} < \text{a q M2} > \leadsto_{\sigma_2, \text{ buy-c}} < \text{a c q M3} > \leadsto_{\sigma_3, \text{ buy-a}} < \text{a a c q M4} >$$

where `M3` and `M4` are variables of the sort `Money` and the substitutions are $\sigma_2 = \{$`M2` $\mapsto$
`M3`, `M` $\mapsto$ `a q M3`$\}$ and $\sigma_3 = \{$ `M3` $\mapsto$ `q q q M4`, `M` $\mapsto$ `a c M4`$\}$ with the rewrite rules `buy-c` and
`buy-a`, respectively. In the third narrowing step, when we apply the substitution $\sigma_3$, the
two obtained instances of `< a c q M3 >` and the left-hand side of the rewrite rule `buy-a`
are `< a c q q q q M4 >` and `< a c M4 >`, respectively. The two instances are actually equal,
thanks to the commutative property and the equation `change` . Therefore, the rewrite rule
`buy-a` modulo the equational theory is used to obtain the state `< a a c q M4 >` . From
what has been described, it follows that by using narrowing, we can solve the reachability
problem where *St* and *St'* are patterns (terms that may have variables in common) of

the sort `State` such that some conditions are satisfied, and $R, E$ are the rewrite rules and equations in the specification.

### Par-Maude-NPA

Par-Maude-NPA (*Do et al., 2022*) has been developed in order to improve the running performance of Maude-NPA by the parallel mechanism. Maude-NPA basically uses a breadth-first search (BFS) to explore the state space backwardly from a given insecure attack pattern. Given a set of states in layer $l$, for each state in the set, the backward narrowing can be performed independently to obtain its successor states in layer $l + 1$. Par-Maude-NPA makes use of this point to parallelize the backward narrowing tasks for states at the same layer, so that successor states are generated in parallel at each layer. Generally speaking, the breadth-first search in Maude-NPA is transformed into a parallel breadth-first search in Par-Maude-NPA without altering the number or form of the states in the state space. Par-Maude-NPA uses a master-worker model with one master and multiple workers, where the master distributes (or assigns) jobs to each worker in a well-balanced way. The tool uses a shared cache maintained by the master and a local cache maintained by each worker to avoid making unnecessary duplications of jobs.

## HYBRID POST-QUANTUM TLS 1.2

Figure 1 depicts the exchanged messages in a full handshake of the Hybrid PQ TLS protocol. In this figure, * indicates that the message is not sent unless client authentication is requested. The hybrid key exchange mechanism in the proposed protocol directly impacts on ClientHello, ServerHello, ServerKeyExchange, and ClientKeyExchange messages.

Let $C$ and $S$ denote a client and a server, respectively. When $C$ wants to start a new session with $S$, $C$ starts by sending a ClientHello message to $S$. This message in the full handshake mode consists of the protocol version, a random number, an empty session ID, a list of cipher suites (cryptographic algorithms supported by $C$) in order of $C$'s preference, and a set of post-quantum KEM parameters (including both the name of KEM and its parameters) supported by $C$. When receiving the ClientHello message, $S$ responds to $C$ with a ServerHello message, containing the protocol version, a random number, a non-empty session ID, and a selected cipher suite. $S$ then sends his/her digital certificate (via a Certificate message) and a ServerKeyExchange message to $C$. The ServerKeyExchange message includes the ECDH & KEM public keys and a signature over the two public keys together with the two random numbers previously sent in the Hello messages (*i.e.*, ClientHello and ServerHello messages) signed by $S$'s long-term private key. In the case when client authentication is requested, which is optional, a CertificateRequest message will also be sent. $S$ then sends a ServerHelloDone message to $C$, informing the completion of the hello phase on the server side.

When receiving the ServerHelloDone message, $C$ replies to $S$ with a ClientKeyExchange message, consisting of the client's ECDH public key and the KEM ciphertext (generated by the algorithm `Encaps`, see the "Protocol Formal Specification in Maude-NPA" section). Before that, however, if $S$ has previously sent a CertificateRequest message, meaning that client authentication was requested, $C$ must send their certificate (a Certificate message)

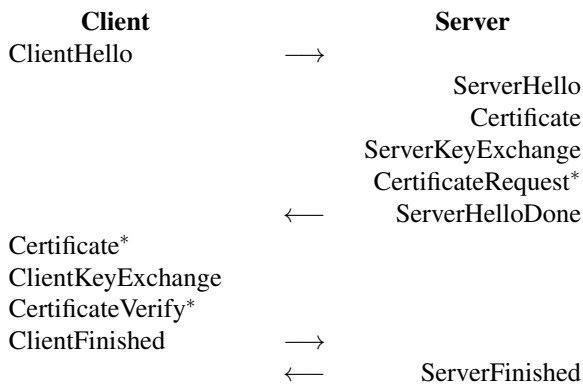

**Figure 1** **Messages exchanged in a full handshake of the hybrid post-quantum TLS protocol (*Campagna & Crockett, 2021*).** An asterisk (∗) indicates that the message is not sent unless client authentication is requested.

first. Furthermore, if client authentication was requested, following the ClientKeyExchange message, $C$ will then send a CertificateVerify message, that is a signature over all handshake messages exchanged so far signed by $C$'s long-term private key. Finally, $C$ sends a ClientFinished message, whose content is a hash of all handshake messages exchanged so far encrypted by the negotiated handshake key.

When receiving the ClientFinished message, $S$ validates it and sends a ServerFinished message. Once $C$ receives the ServerFinished message, it also has to validate that the message is correct. After that, both sides can securely exchange messages by encrypting them under the negotiated handshake key.

## PROTOCOL FORMAL SPECIFICATION IN MAUDE-NPA

Through this section, we present how we formally specify the Hybrid PQ TLS protocol in Maude-NPA. Before going into detail about the protocol formal specification, we first describe the *strand* notation through a running example to see how *strands* can be used to model the protocol execution.

### Formal specification by strands

Maude-NPA uses *strands* (*Thayer, Herzog & Guttman, 1998*) to specify the execution of the protocol under analysis as well as the capabilities of the intruder. Each strand is a sequence of positive and negative messages and possibly with some unique fresh data as follows:

$$:: r_1, \ldots, r_k :: [+(msg_1), -(msg_2), \ldots, -(msg_i) \mid +(msg_{i+1}), \ldots]$$

where $r_1, \ldots, r_k$ denote unique fresh data consumed by the principal performing the strand (they would be, for example, secret keys, unique random numbers, and session identifiers). A positive message $+(msg)$ and a negative message $-(msg)$ denote sending and receiving the message $msg$, respectively. The vertical bar is used to distinguish between present and future when the strand appears in a state description. Messages appearing before the bar

were sent/received in the past, while messages appearing after the bar will be sent/received in the future. Let us consider a classical security protocol, namely Needham-Schroeder-Lowe Public Key (NSLPK) protocol (*Lowe, 1995*), to illustrate how strands would be used to specify the protocol execution. The protocol consists of the following three exchange messages:

(1) $P \rightarrow Q : pk(Q, P; N_P)$
(2) $Q \rightarrow P : pk(P, N_P; N_Q; Q)$
(3) $P \rightarrow Q : pk(Q, N_Q)$

where $P$ and $Q$ denote two principal identifiers, $N_P$ and $N_Q$ are nonces (unguessable values) generated by $P$ and $Q$, respectively, and $pk(P, m)$ denotes the encryption of message $m$ by the public key of $P$. The three messages exchanged can be explained as follows. $P$ first generates a nonce $N_P$ and sends it together with $P$'s identifiers encrypted by $Q$'s public key to $Q$. Upon receiving that message, $Q$ decrypts it and obtains a nonce. $Q$ generates a new nonce $N_Q$, encrypts the nonce received, the new nonce, and $Q$'s identifiers under $P$'s public key, and sends the result back to $P$. Upon receiving the response message, $P$ decrypts it, checking if one of the nonces is exactly the one that $P$ has sent before in the first message. If that is the case, $P$ responds to $Q$ with the other nonce encrypted under $Q$'s public key.

Maude-NPA provides some built-in sorts, such as the sort `Msg` that represents messages and the sort `Fresh` that is used to identify terms that must be unique. To model the protocol in Maude-NPA, we first introduce some new sorts, such as `Name` and `Nonce` to distinguish principal names and nonces, respectively. The two sorts are subsorts of the sort `Msg`, which are declared as follows:

```
sorts Name Nonce .
subsort Name Nonce < Msg .
```

The nonce constructor is specified by the following operator:

```
op n : Name Fresh -> Nonce [frozen] .
```

where the `frozen` attribute is attached following the Maude-NPA convention, which is necessary to tell Maude not to attempt to apply rewriting/narrowing at the arguments of this operator. Suppose that `P` and `r` respectively are variables of the sorts `Name` and `Fresh`, then `n(P,r)` denotes the nonce generated by `P`, where `r` guarantees its uniqueness. We also declare the public encryption operator as follows:

```
op pk : Name Msg -> Msg [frozen] .
```

Let `Q` and `N` be variables of the sorts `Name` and `Nonce`, respectively. The strand specifying the protocol execution from the `P` side is then defined as follows:

```
:: r ::
[ nil | +(pk(Q, P ; n(P,r))), -(pk(P, n(P,r) ; N ; Q)), +(pk(Q, N)), nil ]
```

The strand says that initially, given a unique fresh `r`, `P` consumes it to generate the nonce `n(P,r)` and sends the nonce to `Q` together with `P`'s identifiers encrypted by `Q`'s public key. When `P` receives another message whose content is `n(P,r)`, another nonce `N`, and `Q`'s identifiers encrypted under `P`'s public key (which can be checked by using their secret key to decrypt the received ciphertext), `P` replies to `Q` the new nonce `N` encrypted under `Q`'s public key. Recall that `:: r::` denotes the unique fresh `r` which is consumed by the first

output message of the strand and all messages are put after the vertical bar following the Maude-NPA convention because the vertical bar is irrelevant when specifying the protocol execution. Note also that ; is the infix operator of message concatenation, which has signature declaration as follows:

```
op _;_ : Msg Msg -> Msg [frozen] .
```

In the same manner, the strand that describes how the protocol is executed from the $Q$ side can be specified as follows:

```
:: r ::
[ nil | -(pk(Q, P ; N)), +(pk(P, N ; n(Q,r) ; Q)), -(pk(Q, n(Q,r))), nil ]
```

Strands are also used for the specification of the intruder capabilities. But in this case, such an intruder strand is limited to be in the form of a sequence of negative messages (possibly empty) followed by one positive message combining all previous variables under a function symbol. For example, the intruder's capability in encrypting any message by any public key is specified as follows:

```
:: nil :: [ nil | -(M), +pk(P,M), nil ]
```

where M is a variable of the sort Msg . Precisely, the strand says that if the intruder has learned the message M , then the intruder can encrypt it by the public key of any principal (P ).

## Protocol formal specification

This section describes the Maude-NPA formal specification of the Hybrid PS TLS protocol. We start with the definition of KEMs and how to model them in Maude-NPA.

### *KEM specification*

A key encapsulation mechanism is a tuple of algorithms (KeyGen , Encaps , Decaps) along with a finite key space $\mathcal{K}$:

- KeyGen () $\rightarrow$ $(pk, sk)$: A probabilistic *key generation* algorithm, which generates a public & secret key pair $pk$ & $sk$.
- Encaps $(pk)$ $\rightarrow$ $(c, k)$: A probabilistic *encapsulation* algorithm, which takes as input a public key $pk$ and generates an encapsulation (or ciphertext) $c$ and a shared key $k \in \mathcal{K}$.
- Decaps $(c, sk)$ $\rightarrow$ $k$: A deterministic *decapsulation* algorithm, which takes as inputs a ciphertext $c$ and a secret key $sk$, and returns as output a shared key $k \in \mathcal{K}$.

   A KEM is $\epsilon$- *correct* if for all $(pk, sk) \leftarrow$ KeyGen () and $(c, k) \leftarrow$ Encaps $(pk)$, it holds that: Pr[ Decaps $([c, sk \neq k] \leq \epsilon$

   Intuitively, $\epsilon$ is the probability such that Decaps cannot recover the correct shared key $k$. In most cases, $\epsilon$ would be negligible, and when $\epsilon = 0$, we say the KEM is *correct*. In this article, we only consider the case that all KEMs are *correct*, which means that Encaps and Decaps always correctly return the same shared key $k$. Idealizing assumptions like that are typically necessary when conducting security analysis in the symbolic model. Furthermore, because a Decaps -failure probability is really small, typically almost 0, we are not over-idealizing when omitting such Decaps -failure cases. For example, with Kyber, that failure probability is below $2^{-140}$ (*Bos et al., 2018*), *i.e.*, Kyber is $\epsilon$- *correct* with $\epsilon < 2^{-140}$.

To model KEMs in Maude-NPA, we first introduce sorts `PqSk` , `PqPk` , `Cipher` , and `PqKey` to represent secret keys, public keys, encapsulations, and shared keys, respectively. The constructor of a secret key is specified by the following operator:

```
op pqSk : Name Fresh -> PqSk [frozen] .
```

The argument of the sort `Fresh` provides the uniqueness of secret keys, while the argument of the sort `Name` is not strictly necessary, but it is convenient for identifying who is the owner of a given key.

The `Decaps` algorithm is straightforwardly declared as follows:

```
op decap : Cipher PqSk -> PqKey [frozen] .
```

Unlike `Decaps` , it is a bit tricky to specify the `KeyGen` and `Encaps` procedures because they are probabilistic algorithms. For each of the two procedures, we add an argument of the sort `PqSk` to make them become deterministic procedures. With the `Encaps` procedures, we declare two separate Maude operators: `encapCipher` and `encapKey` that return the ciphertext $c$ and the key $k$, respectively. We declare the following operators:

```
op pqPk        : PqSk       -> PqPk    [frozen] .
op encapCipher : PqPk PqSk -> Cipher [frozen] .
op encapKey    : PqPk PqSk -> PqKey  [frozen] .
```

The first operator models the `KeyGen` procedure, while the two others model the `Encaps` procedure. The algebraic properties of KEMs are then specified by means of equations as follows:

```
op $pqKey : PqSk PqSk -> PqKey [frozen] .
eq encapKey(pqPk(S:PqSk), S2:PqSk) = $pqKey(S:PqSk, S2:PqSk) [variant] .
eq decap(encapCipher(pqPk(S:PqSk),S2:PqSk), S:PqSk)
   = $pqKey(S:PqSk, S2:PqSk) [variant] .
```

where the `variant` attribute denotes that the two equations are not regular Maude equations used for simplification, but are equations used for variant-based equational unification (_Escobar, Sasse & Meseguer, 2010_). Here we introduce one more operator, namely `$pqKey`, which is necessary for specifying that the rewritings of `encapKey` and `decap` (`Encaps` and `Decaps` steps) on proper arguments will result in the same key. The first equation can be straightforwardly comprehended. The second equation states that given an encapsulation _en_ and a secret key _sk_, a principal can perform `Decaps` (_en, sk_) to get the proper shared key only if _en_ is the result of `Encaps` when taking as input the public key associated with the secret key _sk_.

### ECDH and key calculation specification

To model ECDH in Maude-NPA, we first introduce the following sorts:

```
sort Scalar Point ECKey .
subsort Point < ECKey .
```

The sort `Point` represents points on the curve, which serve as ECDH public keys. The sort `Scalar` and `ECKey` represent the secret keys and shared keys, respectively. We then declare the following operators:

```
op p   : -> Point .
op scl : Name Fresh    -> Scalar [frozen] .
op gen : Point Scalar -> Point  [frozen] .
op _*_ : Scalar Scalar -> Scalar [frozen assoc comm] .
```

The constant `p` denotes a point generator on the curve, which is publicly known by all participants including the intruder. The operator `scl` takes as inputs a principal name and a fresh, and outputs a scalar, which serves as a secret key. The operator `gen` takes as inputs a point and a (secret) scalar and outputs another point. Concretely, when the first argument is a point generator, the result is a public key, which is used to send to the other peer; and when the first argument is a public key received from the opposite peer, the result is a shared key. The last operator is the associative-commutative multiplication operation on scalars, thanks to the Maude attributes `assoc` and `comm`. The algebraic property of ECDH is then specified by the following equation:

```
eq gen(gen(P:Point, S:Scalar), S2:Scalar)
   = gen(P:Point, S:Scalar * S2:Scalar) [variant] .
```

Sorts `PreMasterSecret` and `MasterSecret` are introduced to represent premaster secrets and master secrets, respectively, in the protocol key calculation. We then model their calculations with the following operators:

```
op pms : ECKey PqKey -> PreMasterSecret [frozen] .
op ms  : PreMasterSecret Rand Rand Point Cipher -> MasterSecret [frozen] .
```

where `Rand` is the sort representing random numbers generated by clients and servers. A pre-master secret is the concatenation of an ECDH shared secret and a PQ KEM shared secret. Whereas, a master secret is computed by the pseudorandom function (PRF) taking as inputs a pre-master secret and a seed, which is the combination of the two random numbers previously exchanged in the two Hello messages and the ECDH public key & PQ encapsulation previously sent in the ClientKeyExchange message.

### *Honest principal specification*

We use three operators `rd`, `sess`, and `cert` that serve as functions to produce random numbers, session IDs, and digital certificates, respectively. We also define operators `sig` and `enc` reflecting the signature and symmetric encryption functions. All of them are as follows:

```
op rd   : Name Fresh -> Rand [frozen] .
op sess : Name Fresh -> Session [frozen] .
op cert : Name -> Cert [frozen] .
op sig  : Name Msg -> Msg [frozen] .
op enc  : MasterSecret Msg -> Msg [frozen] .
```

Given a server `S`, a message `M`, and a master-secret `MS`, `enc(MS,M)` denotes the ciphertext obtained by encrypting `M` by `MS`, while `sig(S,M)` denotes the signature over message `M` signed by the long-term private key of server `S`. By using the principal name as an input argument for the signature sign function instead of the private key (for example, `sig(priKey(S),M)`), we implicitly associate a long-term private key with its owner's name. For the sake of simplicity and for reducing the size of the state space, `cert(S)` is used to denote the digital certificate of the server `S`, while in fact, the certificate must contain information about the trusted certificate authority and the public key of `S`. By using that simplification form, we explicitly associate a certificate with its owner's name and ignore the case when the intruder tries to fake a certificate.

Let `r1`, `r2`, and `r3` be variables of the sort `Fresh`; `C` and `S` be variables of the sort `Name`; `N` and `SS` be variables of the sorts `Rand` and `Session`, respectively; `PK1` and `PK2` be variables

of the sorts `Point` and `PqPk`, respectively. The execution of the protocol of Fig. 1 up to the ClientFinished message from a client's side is specified as follows:

```
:: r1,r2,r3 ::
[ nil |
 +(ch ; rd(C,r1)),
 -(sh ; N ; SS),
 -(sc ; cert(S)),
 -(ske ; PK1 ; PK2 ; sig(S, PK1 ; PK2 ; rd(C,r1) ; N)),
 +(cke ; gen(p,scl(C,r2)) ; encapCipher(PK2, pqSk(C,r3))),
 +(cf ; enc(ms(pms(gen(PK1,scl(C,r2)), encapKey(PK2, pqSk(C,r3))),
             rd(C,r1), N, gen(p,scl(C,r2)), encapCipher(PK2, pqSk(C,r3))),
     (ch ; rd(C,r1)) ++
     (sh ; N ; SS) ++
     (sc ; cert(S)) ++
     (ske ; PK1 ; PK2 ; sig(S, PK1 ; PK2 ; rd(C,r1) ; N)) ++
     (cke ; gen(p,scl(C,r2)) ; encapCipher(PK2, pqSk(C,r3))))
  ),
nil ]
```

where `ch`, `sh`, `sc`, `ske`, `cke`, and `cf` are constants of the sort `Msg`, which are acronyms of ClientHello, ServerHello, Server Certificate, ServerKeyExchange, ClientKeyExchange, and ClientFinished. The `++` operator denotes message concatenation. The strand says that the client `C` starts a new connection with the server `S` by sending a ClientHello message with a random number denoted by `rd(C,r1)`. When the client receives a ServerHello message, a valid server Certificate message, and a valid ServerKeyExchange message with ECDH and PQ public keys, *i.e.*, `PK1` and `PK2`, the client will send a ClientKeyExchange message with the client's ECDH public key exchange and the encapsulation computed with `PK2` as one of the inputs. Then, the client also sends a ClientFinished message, that is the concatenation (denoted by `++`) of all messages exchanged so far encrypted under the master secret. Note that for the sake of simplicity and for reducing the size of the state space, we use master secrets as symmetric keys for encryption of Finished messages. However, the correct protocol design states such a symmetric key must be computed by the PRF function taking as inputs the master secret and the two random numbers in the two Hello messages sent in the same session. Moreover, the ServerHelloDone message and some components of the Hello messages, such as protocol versions, cipher suites, and lists of cipher suites, are excluded. We also suppose that client authentication is not requested. Note also that the use of the concatenation operator `++` is not strictly necessary. For example, from the definition of `++`, it follows that `(sh ; N ; SS) ++ (sc ; cert(S))` is rewritten to `(sh ; N ; SS ; sc ; cert(S))`. Therefore, `++` can be omitted without posing any technical problems. However, we define and keep on using it because we want to show each message separately for ease of understanding from readers.

In the same manner, we specify the protocol execution from a server's side. The strand below specifies the server execution up to the ClientFinished message:

```
:: r1,r2,r3,r4 ::
[ nil |
 -(ch ; N),
 +(sh ; rd(S,r1) ; sess(S,r2)),
 +(sc ; cert(S)),
 +(ske ; gen(p,scl(S,r3)) ; pqPk(pqSk(S,r4)) ;
     sig(S, gen(p,scl(S,r3)) ; pqPk(pqSk(S,r4)) ; N ; rd(S,r1))),
 -(cke ; PK1 ; CP),
 -(cf ; enc(ms(pms(gen(PK1,scl(S,r3)), decap(CP, pqSk(S,r4))),
             N, rd(S,r1), PK1, CP),
         (ch ; N) ++
```

```
          (sh  ;  rd(S,r1)  ;  sess(S,r2)) ++
          (sc  ;  cert(S)) ++
          (ske  ;  gen(p,scl(S,r3))  ;  pqPk(pqSk(S,r4))  ;
             sig(S,  gen(p,scl(S,r3))  ;  pqPk(pqSk(S,r4))  ;  N  ;  rd(S,r1))) ++
          (cke  ;  PK1  ;  CP))
       ),
nil ]
```

where `CP` is a variable of the sort `Cipher`. Note that in both strands, we omit the ServerFinished message because they are too long to show all. The complete specification is publicly available at https://doi.org/10.5281/zenodo.7919153.

### *Intruder capabilities*

Maude-NPA uses the standard Dolev-Yao intruder model (*Dolev & Yao, 1983*). The intruder is given the capability of fully controlling the network, in particular, the intruder can intercept and learn information from any message in the network; fake and synthesize messages based on the gleaned information; and impersonate some principal to send the faking message to some other. Let `X` and `Y` be variables of the sort `Msg`, we first specify the intruder's capabilities in concatenating and de-concatenating messages by the following three strands:

```
:: nil :: [ nil | -(X), -(Y), +(X ; Y), nil ] &
:: nil :: [ nil | -(X ; Y), +(X), nil ] &
:: nil :: [ nil | -(X ; Y), +(Y), nil ] &
```

The intruder may consume a fresh to generate by themself a random number and a scalar serving as an ECDH secret key, which is specified as follows:

```
:: r :: [ nil | +(rd(i,r)), nil ] &
:: r :: [ nil | +(scl(i,r)), nil ] &
```

where `i` is a constant denoting the intruder.

With KEMs, if the intruder has learned some public key `PK2`, secret key `SK`, and encapsulation `CP`, then the intruder can derive some appropriate encapsulations and keys as follows:

```
:: nil :: [ nil | -(PK2), -(SK), +(encapCipher(PK2,SK)), nil ] &
:: nil :: [ nil | -(PK2), -(SK), +(encapKey(PK2,SK)), nil ] &
:: nil :: [ nil | -(CP), -(SK), +(decap(CP,SK)), nil ] &
```

With ECDH, an important assumption we suppose is that the intruder can break the key exchange security by utilizing the power of quantum computation. That is, if the intruder knows the two ECDH public keys exchanged between a client and a server, then the intruder can derive the shared secret key. This capability is specified by the following strand:

```
:: nil :: [ nil | -(gen(p,S)), -(gen(p,S2)), +(gen(p,S * S2)), nil ]
```

where `S` and `S2` are variables of `Scalar`.

The complete specification of the intruder capabilities includes some more strands, which are omitted showing here. Again, readers can find them at https://doi.org/10.5281/zenodo.7919153.

## Checking the specification

To increase trust in the correctness of what has been formally specified, a state pattern is defined proving that it is possible to successfully complete a protocol handshake between

two principals. This must be fulfilled, otherwise, the analysis coming later would be meaningless. Let `c` and `s` be two constants of the sort `Name`, denoting arbitrary principals (implicitly stand for a client and a server). The state pattern is defined as a Maude-NPA `ATTACK-STATE` as follows:

```
eq ATTACK-STATE(10) =
:: r1,r2,r3 ::
[ nil,
 +(ch ; rd(c,r1)),
 -(sh ; N ; SS),
 -(sc ; cert(s)),
 -(ske ; PK1 ; PK2 ; sig(s, PK1 ; PK2 ; rd(c,r1) ; N)),
 +(cke ; gen(p, scl(c,r2)) ; encapCipher(PK2, pqSk(c,r3))),
 +(cf ; enc(ms(pms(gen(PK1, scl(c,r2)), encapKey(PK2, pqSk(c,r3))),
    rd(c,r1), N, gen(p, scl(c,r2)), encapCipher(PK2, pqSk(c,r3))),
   (ch ; rd(c,r1)) ++
   (sh ; N ; SS) ++
   (sc ; cert(s)) ++
   (ske ; PK1 ; PK2 ; sig(s, PK1 ; PK2 ; rd(c,r1) ; N)) ++
   (cke ; gen(p, scl(c,r2)) ; encapCipher(PK2, pqSk(c,r3))))),
 -(sf ; ... )
 | nil ]
|| empty
|| nil    || nil    || nil
[nonexec] .
```

where `10` in `ATTACK-STATE(10)` denotes the attack pattern number in order to distinguish from other attacks and ... denotes that some terms of the sort `Msg`, which are components of the ServerFinished message, are omitted for the sake of simplicity. The attack pattern consists of five sections separated by the symbol ||. The second section is the intruder knowledge, which is empty in this case. The last three sections are set to `nil` denoting empty because they are irrelevant in this case. The first section is the set of strands expected to appear in the attack, which says that client `c` has completed the protocol handshake with server `s`. Maude-NPA returns a solution for the attack pattern, thereby confirming that it is possible to successfully complete the protocol handshake between two principals.

## ANALYSIS EXPERIMENTS

Three experiments analyze the following three properties for the protocol with each of Maude-NPA and Par-Maude-NPA: (1) the secrecy property of the ECDH shared secret key established between two honest principals, (2) the secrecy property of the PQ KEM shared secret key established between two honest principals, and (3) the authentication property, *i.e.*, if *A* has completed the protocol execution apparently with *B*, then *B* is indeed the principal who has communicated with *A*.

### Secrecy property of ECDH shared secret key

The first property we would like to check is whether the intruder can glean the ECDH shared secret key when a client and a server have completed the handshake. To check that property, we specify another Maude-NPA attack pattern, *i.e.*, `ATTACK-STATE(0)`, that is identical to the `ATTACK-STATE(10)` shown above except for the intruder knowledge section, which is changed from `empty` to the following:

```
gen(PK1, scl(c,r2)) inI, empty
```

Recall that `PK1` denotes the ECDH public key that client `c` receives from server `s`, while `scl(c,r2)` denotes the ECDH secret key of the client. The symbol `_inI` indicates what messages the intruder knows, while the symbol `_!inI` indicates what messages the intruder does not yet know (but will know in the future). The attack pattern says that when a client has completed the handshake with a server in which the server sent their ECDH public key, *i.e.*, `PK1`, to the client and the client sent their ECDH public key, *i.e.*, `gen(p, scl(c,r2))` to the server, then the intruder can learn the corresponding ECDH shared key denoted by `gen(PK1, scl(s,r2))`.

We have checked the attack pattern with both the original Maude-NPA and Par-Maude-NPA. Par-Maude-NPA found a counterexample for the attack pattern in about 1,722 h (nearly 72 days), meaning that the intruder can learn the ECDH shared secret key established between the honest client and the honest server. That counterexample state contains four sections: (1) state ID, (2) set of current protocol and intruder strands, (3) intruder knowledge, and (4) sequence of messages. To understand how the attack can happen, we show (4)—the message sequence leading to the counterexample as follows:

```
1   +(ch ; rd(c,#3:Fresh)),
2   -(ch ; rd(c,#3:Fresh)),
3   +(sh ; rd(s,#4:Fresh) ; sess(s,#6:Fresh)),
4   +(sc ; cert(s)),
5   +(ske ; gen(p, scl(s,#0:Fresh)) ; pqPk(pqSk(s,#2:Fresh)) ;
sig(s, gen(p, scl(s,#0:Fresh)) ; pqPk(pqSk(s,#2:Fresh)) ;
rd(c,#3:Fresh) ; rd(s,#4:Fresh))),
6   -(ske ; ...),
7   +(gen(p, scl(s,#0:Fresh)) ; pqPk(pqSk(s,#2:Fresh)) ;
sig(s, gen(p, scl(s,#0:Fresh)) ; pqPk(pqSk(s,#2:Fresh)) ;
rd(c,#3:Fresh) ; rd(s,#4:Fresh))),
8   -(sh ; rd(s,#4:Fresh) ; sess(s,#6:Fresh)),
9   -(sc ; cert(s)),
10  -(ske ; ...),
11  +(cke ; gen(p, scl(c,#1:Fresh)) ;
encapCipher(pqPk(pqSk(s,#2:Fresh)), pqSk(c,#5:Fresh))),
12  -(cke ; ...),
13  +(gen(p, scl(c,#1:Fresh)) ;
encapCipher(pqPk(pqSk(s,#2:Fresh)),pqSk(c,#5:Fresh))),
14  -(gen(p, scl(c,#1:Fresh)) ;
encapCipher(pqPk(pqSk(s,#2:Fresh)),pqSk(c,#5:Fresh))),
15  +(gen(p, scl(c,#1:Fresh))),
16  -(gen(p, scl(s,#0:Fresh)) ;
pqPk(pqSk(s,#2:Fresh)) ; sig(s, gen(p, scl(s,#0:Fresh)) ;
pqPk(pqSk(s,#2:Fresh)) ; rd(c,#3:Fresh) ; rd(s,#4:Fresh))),
17  +(gen(p, scl(s,#0:Fresh))),
18  -(gen(p, scl(s,#0:Fresh))),
19  -(gen(p, scl(c,#1:Fresh))),
20  +(gen(p, scl(s,#0:Fresh) * scl(c,#1:Fresh))),
21  +(cf ; ...),
22  -(cke ; gen(p, scl(c,#1:Fresh)) ;
encapCipher(pqPk(pqSk(s,#2:Fresh)), pqSk(c,#5:Fresh))),
23  -(cf ; ...),
24  +(sf ; ...),
25  -(sf ; ...)
```

We insert line numbers on the left side for easy reference in the following explanation. Client `c` starts sending a ClientHello message to server `s` (at line 1), where `#3:Fresh` denote a unique fresh. Server `s` receives that message (at line 2), and then sends consecutively to `c` a ServerHello message (at line 3), a Certificate message (at line 4), and a ServerKeyExchange message (at line 5). The intruder grasps the ServerKeyExchange message, and by applying the de-concatenation rules twice, they learn the ECDH public key (`gen(p, scl(s,#0:Fresh))`) sent in that message (at lines 6–7 and 16–17). We use ... to omit some components of

a message, which is too long to show fully. When `c` receives the three messages from `s` (at lines 8–10), `c` sends to `s` a ClientKeyExchange message (at line 11). The intruder again grasps that message, and by applying the de-concatenation rules twice, they learn the ECDH public key (`gen(p, scl(s,#1:Fresh))`) sent in that message (at lines 12–15). From the two ECDH public keys learned (at lines 18–19), by applying the rule

```
:: nil :: [ nil | -(gen(p,S)), -(gen(p,S2)), +(gen(p,S * S2)), nil ]
```

the intruder learns the ECDH shared secret between `c` and `s` (at line 20). `c` then sends a ClientFinished message to `s` (at line 21). Upon receiving the ClientKeyExchange and ClientFinished messages (at lines 22–23), `s` sends back to `c` a ServerFinished message (at line 24). The last line 25 denotes the reception of that ServerFinished message on the client side. The handshake is now completed, in which the ECDH shared secret is learned by the intruder.

The counterexample was found by Par-Maude-NPA at depth 12. It means that the found state is located at depth 12 from the root of the backward reachability tree constructed by the tool. In this tree, the root is the attack pattern, and a child node is connected to its parent node by an action of an honest principal or the intruder. With Maude-NPA, it did not find the counterexample in the same amount of time T. The detail of the experimental results is discussed in Section 'Experimental results'.

## Secrecy property of PQ KEM shared secret key

Another attack pattern namely `ATTACK-STATE(1)` is defined as the counterpart of `ATTACK-STATE(0)`, checking whether the intruder can learn the PQ KEM shared secret key established between the client and the server. If it is possible, then they can indeed derive the pre-master secret and the master secret from both PQ KEM and ECDH shared secrets learned; otherwise, the secrecy of the master secret is guaranteed. Therefore, the secrecy property of handshake keys is checked through this attack pattern. `ATTACK-STATE(1)` is identical to `ATTACK-STATE(10)` except for the intruder knowledge section, which now becomes as follows:

```
encapKey(PK2, pqSk(c,r3)) inI, empty
```

`ATTACK-STATE(1)` says that when a client has completed the protocol handshake with a server in which the server sent the PQ KEM public key denoted by `PK2` to the client and the client sent the PQ KEM ciphertext (or encapsulation) denoted by `encapCipher(PK2, pqSk(c,r3))` to the server, then the intruder cannot learn the PQ KEM shared key denoted by `encapKey(PK2, pqSk(c,r3))`.

Par-Maude-NPA did not find any counterexample up to depth 12, confirming that the intruder cannot learn the PQ KEM shared secret established between the client and the server up to the bounded depth. Together with the previous analysis experiment, it follows that even though the intruder can learn the ECDH shared secret, the intruder cannot learn the master secret. We leave a detailed discussion about the analysis result in Section 'Experimental results'.

## Authentication property

The next property we would like to check is the *authentication* property, *i.e.*, if client *A* has completed the handshake apparently with server *B*, then *B* is indeed the principal who has communicated with *A*. We define the following ATTACK-STATE(2) :

```
eq ATTACK-STATE(2) =
:: r1,r2,r3 ::
[ nil,
 +(ch ; rd(c,r1)),
 -(sh ; N ; SS),
 -(sc ; cert(s)),
 -(ske ; PK1 ; PK2 ; sig(s, PK1 ; PK2 ; rd(c,r1) ; N)),
 +(cke ; gen(p,pt(c,r2)) ; encapCipher(PK2, pqSk(c,r3))),
 +(cf ; enc(ms(pms(gen(PK1, pt(c,r2)), encapKey(PK2, pqSk(c,r3))),
    rd(c,r1),N,gen(p,pt(c,r2)),encapCipher(PK2, pqSk(c,r3))),
   (ch ; rd(c,r1)) ++
   (sh ; N ; SS) ++
   (sc ; cert(s)) ++
   (ske ; PK1 ; PK2 ; sig(s, PK1 ; PK2 ; rd(c,r1) ; N)) ++
   (cke ; gen(p,pt(c,r2)) ; encapCipher(PK2, pqSk(c,r3))))),
 -(sf ; ... )
 | nil ]
|| empty
|| nil
|| nil
|| never(
  :: r',r1',r2',r3' ::
  [ nil |
   -(ch ; rd(c,r1)),
   +(sh ; N ; SS),
   +(sc ; cert(s)),
   +(ske ; PK1 ; PK2 ; sig(s, PK1 ; PK2 ; rd(c,r1) ; N)),
   -(cke ; gen(p,pt(c,r2)) ; encapCipher(PK2, pqSk(c,r3))),
   -(cf ; enc(ms(pms(gen(PK1,pt(c,r2)), encapKey(PK2, pqSk(c,r3))),
      rd(c,r1),N,gen(p,pt(c,r2)),encapCipher(PK2, pqSk(c,r3))),
     (ch ; rd(c,r1)) ++
     (sh ; N ; SS) ++
     (sc ; cert(s)) ++
     (ske ; PK1 ; PK2 ; sig(s, PK1 ; PK2 ; rd(c,r1) ; N)) ++
     (cke ; gen(p,pt(c,r2)) ; encapCipher(PK2, pqSk(c,r3))))),
   +(sf ; ... ),
   nil ]
  & STR:StrandSet
  || IK:IntruderKnowledge)
[nonexec] .
```

Recall that ... denotes the omission of some terms of sort Msg , which are components of the ServerFinished message. The intruder knowledge section now becomes empty , meaning that there is no constraint about the intruder knowledge, while the last section now is a never pattern. This attack pattern defines a state in which client c has completed the handshake, apparently with server s , by sending a ClientFinished message to s and c has received back another valid ServerFinished message, but s has actually not executed such a server instance with c (denoted by the never pattern in the last section of the attack). Maude-NPA and Par-Maude-NPA did not find any counterexample up to depths 17 and 18, respectively. Next, we report the detailed experimental results of the three analyses in the next subsection.

## Experimental results

We have conducted experiments for the three above-mentioned attack patterns with Maude-NPA and Par-Maude-NPA on a supercomputer that carries 1.5 TB of memory and four 2.8 GHz microprocessors, where each microprocessor has 16 cores. Because we were

**Table 1 Experimental results of the three attack patterns after 1,722h 20m 23s.**

| Attack number | Maude-NPA | | | Par-Maude-NPA | | |
|---|---|---|---|---|---|---|
| | Time (h:m:s) | Depth | Result | Time (h:m:s) | Depth | Result |
| 0 | 1,722:20:23 | H 11 | ∅ | 1,722:20:23 | F 12 | × |
| 1 | 1,722:20:23 | H 11 | ∅ | 1,722:20:23 | H 13 | ∅ |
| 2 | 1,722:20:23 | H 18 | ∅ | 1,722:20:23 | H 19 | ∅ |

only able to use one such supercomputer for our experiments and could have predicted that it would take a long time to complete each of the total six experiments, we started the six experiments on the supercomputer simultaneously. With Par-Maude-NPA, we used one master and eight workers with at most 256 GB of memory and 16 microprocessor cores for each experiment. With Maude-NPA, at most 256 GB of memory and 4 microprocessor cores were used for each experiment. This resource allocation is reasonably fair because of 1.5 TB memory ($= 6 \times 256$ GB memory) and $4 \times 16$ cores ($> 3 \times 16$ cores $+ 3 \times 4$ cores), although the OS uses some resources, which should be negligible. Note that Maude-NPA uses one core for each experiment and the six experiments may use a total of about 1.5 TB of memory for about 1,722 h. Table 1 shows the experimental results for the three attack patterns with Maude-NPA and Par-Maude-NPA. The first column denotes the attack number corresponding to each attack pattern. The next three columns denote the running time, the depth at which Maude-NPA reaches, and the result of each analysis, respectively, with Maude-NPA and similarly for Par-Maude-NPA with the last three columns. In the **Result** column, the symbol ∅ denotes no counterexample is found up to a bounded depth; while the symbol × denotes a counterexample is found and the protocol is insecure against the attack. In the **Depth** column, "H $x$" denotes the analysis is still running at depth $x$ and has not been completed, where "H" stands for "Handling". On the other hand, "F $x$" denotes the analysis is completed at depth $x$, where "F" stands for "Finished".

With attack pattern number 0, as T $= 1,722$ h, 20 min, and 23 s (nearly 72 days), Par-Maude-NPA completed the analysis at depth 12 and found a counterexample. We decided to stop all of the six experiments as the time T because the long time had been spent. We report the results of the five other experiments as the same time T. Also with attack pattern number 0, as the same time T, Maude-NPA only reached depth 11 and was handling it. Therefore, no counterexample was found by Maude-NPA in T.

With attack pattern number 1, as the same time T, both Par-Maude-NPA and Maude-NPA did not find any counterexample. Par-Maude-NPA and Maude-NPA were handling depth 13 and depth 11, respectively. Observing the number of reachable states at each depth, we see that it is increased after each layer depth. The number of reachable states at each depth of the three analysis experiments is depicted in Table 2, where the hyphen symbol denotes the data is not available because the analysis with that given attack is not conducted at that given depth. Note that the data shown in Table 2 are exactly the same up to depths 10, 10, and 17 for the three attack patterns 0, 1, and 2, respectively, with Maude-NPA and Par-Maude-NPA.

**Table 2  Number of reachable states at each depth.**

| Depth | Attack 0 | Attack 1 | Attack 2 |
|---|---|---|---|
| 1 | 15 | 15 | 8 |
| 2 | 30 | 30 | 5 |
| 3 | 51 | 56 | 2 |
| 4 | 86 | 101 | 2 |
| 5 | 150 | 165 | 2 |
| 6 | 261 | 262 | 2 |
| 7 | 435 | 408 | 2 |
| 8 | 695 | 610 | 2 |
| 9 | 1,067 | 871 | 2 |
| 10 | 1,597 | 1,216 | 4 |
| 11 | 2,374 | 1,757 | 7 |
| 12 | 3,679 | 2,634 | 10 |
| 13 | – | – | 19 |
| 14 | – | – | 40 |
| 15 | – | – | 95 |
| 16 | – | – | 246 |
| 17 | – | – | 602 |
| 18 | – | – | 1,355 |

With attack pattern number 2, similarly, both Par-Maude-NPA and Maude-NPA did not find any counterexample, and were handling at depth 19 and depth 18, respectively, in the time T. The number of reachable states at each depth from 1 to 13 is reasonably small. However, it quickly increases at the deeper depths, especially depths 17 and 18. Therefore, we guess that the experiments would not converge at depth 19 or a bit deeper depth.

The six experiments show that using Par-Maude-NPA helped us to significantly reduce the analysis time in this Hybrid PQ TLS case study. To measure precisely how much Par-Maude-NPA is faster than Maude-NPA, we conduct some more experiments with attack number 0, bounded depths 8 and 9, numbers of workers 8, 12, and 16, and show the results in Table 3. For example, when the bounded depth is 8, we can increase the running performance of Maude-NPA with Par-Maude-NPA as 3.8x, 4.2x, and 4.6x faster than that with Maude-NPA, respectively. The improvement of the running performance of Maude-NPA with Par-Maude-NPA for the bounded depth 9 is better than that for the bounded depth 8 in this case study. That is because the number of states at the bounded depth 9 is larger than that at the bounded depth 8. It says that the more states located at a depth, the more improvement will be obtained by using Par-Maude-NPA. Indeed, the number of states at deeper depths (see Table 2) increases quickly, and then the use of Par-Maude-NPA will achieve a better improvement compared to Maude-NPA in terms of running performance, which was partially shown in these experiments and also in the six experiments described above.

**Table 3  Comparison of Maude-NPA and Par-Maude-NPA in terms of running performance.**

| Attack number | Bounded depth | Maude-NPA Time (h:m:s) | Par-Maude-NPA | | Improvement (how faster) |
|---|---|---|---|---|---|
| | | | #workers | Time (h:m:s) | |
| 0 | 8 | 70:39:03 | 8 | 18:46:37 | 3.8x |
| | | | 12 | 16:54:15 | 4.2x |
| | | | 16 | 15:26:02 | 4.6x |
| 0 | 9 | 304:01:08 | 8 | 57:27:42 | 5.3x |
| | | | 12 | 54:15:12 | 5.6x |
| | | | 16 | 41:38:10 | 7.3x |

# RELATED WORK

Symbolic analysis has been used to analyze a wide range of cryptographic protocols, partially resulting in the modern cryptosystems used today. With the development of post-quantum cryptography, symbolic analysis is once again an efficient approach to their security verification. *Hülsing et al. (2021)* have verified the security of their proposed post-quantum WireGuard (PQ-WireGuard) protocol. This is a quantum-resistant version of the WireGuard protocol (*Donenfeld, 2017*), a lightweight and high-performance VPN protocol. The verification confirms that the protocol enjoys the desired security properties inherited from the WireGuard protocol and also resists attacks using large-scale quantum computers. The verification used the Tamarin prover (*Basin et al., 2017*), which is known as one of the most state-of-the-art formal verification tools for symbolic analysis of cryptographic protocols. Similar to the analysis with Maude-NPA, they have first symbolically modeled the primitives, messages, etc. used in the protocol as function symbols and terms, and then specified the desired security properties. However, unlike Maude-NPA, several commonly used cryptographic primitives are pre-defined as built-in functions in Tamarin, such as the Diffie-Hellman key exchange algorithm, encryption & decryption, hashing, and digital signature, while Maude-NPA leaves all of these definitions to human users. To prove the protocol enjoys the desired properties, they have introduced some auxiliary lemmas, where conjecturing lemmas is known as one of the most intellectual tasks in theorem proving. Additionally, the article has also presented a computational security proof, which gave stronger security guarantees than the symbol proof since probability and complexity are taken into account and fewer idealizing assumptions are made. However, more security properties are verified in the symbolic verification, and more importantly, the symbolic verification is computer-verified.

By using Tamarin, a comprehensive security analysis of TLS 1.3 (*Rescorla, 2018*), precisely, the TLS 1.3 draft 21 release candidate has been presented by *Cremers et al. (2017)*. The security requirements claimed in the draft have been verified with respect to a Dolev-Yao intruder. The analysis has taken into account all possible handshake modes, such as pre-shared key (PSK) based resumption and zero round trip time (0-RTT), which are new mechanisms only available from version 1.3. Similar to the above-mentioned case study by *Hülsing et al. (2021)*, a number of auxiliary lemmas has been used for the security verification through the interactive mode of the tool.

The interactive mode of Tamarin with the employment of some lemmas as mentioned before would be helpful when the property under verification is impossible to be proved in the fully automated mode. Tamarin's verification method is based on constraint solving, and its operation is essentially based on multiset rewriting. Similar to Maude-NPA, the tool can handle an unbounded number of participants and sessions (protocol executions). If the tool terminates, either a counterexample representing an attack is returned or the security property under verification is confirmed. Generally speaking, a Tamarin specification is a state machine, where each state is a multiset of *facts*, and transitions between states are defined by a set of *rules*. Those rules specify the protocol execution, the behavior of honest participants as well as the capabilities of the intruder. A security property is modeled as a trace property, and then Tamarin checks the satisfiability and/or the validity of the property. If it is the case of checking the validity, Tamarin first converts it to checking the satisfiability of the negated formula formalizing the property. Constraint solving is then used to perform an exhaustive, symbolic search for executions with the trace until a satisfying trace is found or no more rewrite rules can be applied. Roughly speaking, the negation of the formula formalizing the validity property in Tamarin corresponds to the attack pattern in Maude-NPA, and the satisfying trace, if found, in Tamarin corresponds to the initial state, if found, in Maude-NPA.

ProVerif (*Blanchet, Cheval & Cortier, 2022*) is another automated tool for symbolically reasoning cryptographic protocols that supports verification with respect to an unbounded number of sessions. A variant of the pi-calculus (*Blanchet, 2016*) is used to model a cryptographic protocol in the presence of a Dolev-Yao intruder (*Dolev & Yao, 1983*), and then ProVerif translates it to a set of Horn clauses. This Horn clause representation makes some abstractions, which is the cost for the support of an unbounded number of sessions. Given a security property that we want to prove, the tool reduces the problem of finding an attack against the property to the derivability of a fact on the Horn clauses representing the protocol execution. If the fact is not derivable from the clauses, the property is proved. On the other hand, if the fact is derivable from the clauses, there may be an attack violating the property under analysis, but it may also be a "false attack", that is the found derivation actually does not correspond to a real attack.

## CONCLUSION

This article has presented a formal analysis of the Hybrid PQ TLS protocol with Maude-NPA and the parallel version of Maude-NPA. The most burdensome problem in the analysis is the big state space generated, which caused a long analysis time for our experiments. This problem, however, is not dedicated to Maude-NPA only, but rather the major drawback of other formal method tools based on exhaustive search. In fact, the running performance of Maude-NPA has significantly improved since several techniques have been proposed and implemented in the tool to reduce the size of the search space, such as generating formal grammars representing terms unreachable from initial states and using a super lazy intruder to delay the generation of substitution instances as much as possible (*Escobar, Meadows & Meseguer, 2008*). Moreover, using the parallel version of Maude-NPA allows

us to obtain a better running performance than the original tool in this formal analysis case study.

To prepare for secure information communication once practical quantum computers become available, security analysis by formal methods is a useful approach to the verification and construction of secure post-quantum cryptosystems. Security verification by Maude-NPA on the one hand is fully automated, that is no manual effort is required once the formal specification and the attack pattern are provided. On the other hand, it may take a quite long time if the state space of the system under verification is huge (*i.e.*, state explosion). One possible way to mitigate the analysis time is by introducing some never patterns serving as auxiliary lemmas to discard some useless reachability branches. This technique has been demonstrated through the case studies by *González-Burgueño et al. (2014)*; *González-Burgueño et al. (2015)*. The technique would be one piece of our future work to investigate on. Besides, we are also interested in formal verifications of post-quantum cryptographic protocols using some interactive approaches that require manual efforts from human users. The CafeOBJ/proof score approach (*Ogata & Futatsugi, 2013*; *Ogata & Futatsugi, 2003*) is such a promising method. We plan to integrate the two approaches to make the best use of the benefits obtained from them and mitigate their weak points.

### Funding

This work was supported by the JST SICORP (No. JPMJSC20C2), the European Regional Development Fund, the Generalitat Valenciana, and the European Union NextGenerationEU. There was no additional external funding received for this study. The funders had no role in study design, data collection and analysis, decision to publish, or preparation of the manuscript.

### Grant Disclosures

The following grant information was disclosed by the authors:
The JST SICORP: JPMJSC20C2.
The European Regional Development Fund.
The Generalitat Valenciana, and the European Union NextGenerationEU.

### Competing Interests

The authors declare that there are no competing interests.

### Author Contributions

- Duong Dinh Tran conceived and designed the experiments, performed the experiments, analyzed the data, performed the computation work, prepared figures and/or tables, authored or reviewed drafts of the article, and approved the final draft.
- Canh Minh Do conceived and designed the experiments, performed the experiments, analyzed the data, performed the computation work, prepared figures and/or tables, authored or reviewed drafts of the article, and approved the final draft.

- Santiago Escobar conceived and designed the experiments, analyzed the data, authored or reviewed drafts of the article, and approved the final draft.
- Kazuhiro Ogata conceived and designed the experiments, analyzed the data, authored or reviewed drafts of the article, and approved the final draft.

## Data Deposition

The complete protocol specification and the attack patterns are available at Zenodo: duongtd23. (2023). Hybrid Post-Quantum TLS formal analysis in Maude-NPA (v1.1). Zenodo. https://doi.org/10.5281/zenodo.7919153.

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
