# Peer review of "Hybrid post-quantum Transport Layer Security formal analysis in Maude-NPA and its parallel version"

_PeerJ Computer Science, doi:10.7717/peerj-cs.1556_

## Round 0.1 · original submission · Major Revisions

While there has been no objection for this article to be accepted eventually, there have been some important comments from the reviewers that would require a major revision. These include: (1) better summarisation of the research problems and novel contributions of this article; (2) experiments using both Maude-NPA and Par-Maude-NPA. As one reviewer mentioned, "the parallel version is more efficient, but the improvement is still limited. Par-Maude-NPA is an existing work and shall not be considered the contribution of this paper. It is unclear why the authors chose 1722 hours as the threshold. What do depths 13 and 19 mean to the protocol? How much confidence can we get from the verification results up to these steps? ". The other reviewer also queried that the focus on Par-Maude-NPA (in some sessions), and why not on TLS in order to focus on the novelty of this article with respect to the workshop paper. The following comments raised by the other reviewer should also been addressed:

My most important criticism has to do with the provided sources.
I have taken a look at the Maude-NPA specification at
https://github.com/duongtd23/PQTLS-MaudeNPA/blob/main/HB-TLS.maude.
I think it would have been better to:
- remove the generic comments, e.g.
"Overwrite this module with the algebraic properties
of your protocol",
- add more informative comments about the code instead,
- include ATTACK-STATE(0), -(1) and -(2) at least as
comments.

·

Basic reporting

The paper meets the expected standards regarding the English used, references, and data.

Experimental design

The experimental design is OK, everything is made available.

Validity of the findings

The validity is basically OK. The paper presents a good overview of Maude and the specification and analysis of TLS. My main concern is that the focus in some sections is on Par-Maude-NPA, showing that it works better then the sequential version. This is weird because this has been already stated in the corresponding paper; in this one I would expect the authors to focus on TLS. That is, the novelty with respect to the workshop paper should be extended.
Moreover, in Lines 643 - 651 you show that Par-Maude-NPA is better than Maude-NPA, but we already knew/expected it. Is TLS good for Par-Maude-NPA? Is it bad but even in that case it works better? Is the performance of Par-Maude-NPA independent of the protocol?

Additional comments

Medium/minor:
- Line 110: you talk about equational attributes, perhaps you should give more details here.
- Line 241: Why it is not a ctor?
- Line 262: I expect it to be assoc. Moreover, in Lines 405-406 you talk about another operator for concatenating, this should be clarified.
- Line 304: you talk here about the symbolic model for the first time. In the introduction you should talk introduce the two possible models and indicate you follow the symbolic one.
- Line 531: I would remove the word "result".
- Line 594: of the sort Msg -> of sort Msg.
- Line 601: analysis -> analyses.
- Line 712: is a big state -> is the big state

Reviewer 2 ·

Basic reporting

This paper presents a case study of modeling and verifying a quantum TLS protocol using Maude. The protocol was proposed by Amazon Web Services and was believed invulnerable under quantum computers. The authors show how such a protocol was modeled and specified using Maude in detail. Then, a comprehensive formal analysis was conducted, and the authors reported that a counterexample was found for the secrecy property and other two properties were verified to hold up to 12 and 18 depth, respectively.

Experimental design

no comment

Validity of the findings

Pros and cons of the paper:

+ The paper addresses an important problem of formally verifying the security of TLS protocol. As an industry-level protocol, TLS's security shall be well-certified. I believe the verification results of the work show the strength of the protocol to some extent.

+ The modeling and verification of the protocol using Maude is non-trivial. In contrast, it took a great effort, such as 72 days, for the experiment on supercomputers. Such efforts should be appreciated. The authors also explain in detail how the protocol is modeled in Maude, which is helpful to other researchers to conduct similar experiments.

+ The paper is well-written and easy to follow. The authors provide many nice details about the modeling of the protocol, which I think is very important and helpful.

However, there are several places that can be improved.

- The research problems and contributions shall be summarized. The work can be well-motivated. Some insights into the problems and the work shall be explicitly summarized.

- The authors conducted experiments using both Maude-NPA and Par-Maude-NPA. I miss the point of doing this. Clearly, the parallel version is more efficient, but the improvement is still limited. Par-Maude-NPA is an existing work and shall not be considered the contribution of this paper. It is unclear to me why the authors chose 1722 hours as the threshold. What do depths 13 and 19 mean to the protocol? How much confidence can we get from the verification results up to these steps?

- Readers may be interested in the counterexample, which I think should be a key highlight of the work. Unfortunately, I didn't find any explanation for it. Maybe it isn't very easy to explain. This important finding deserves a full discussion. If this example has been reported, the paper's contribution is weak.

Some minor comments.

1. In Table 1, all the times are the same for the verification. Is that true, particularly for the case of finding a counterexample?

2. The number of reachable states for each step seems not too many, according to Table 2. Why does it take too much time to verify?

3. Is it possible to improve the verification efficiency by state space reduction via abstraction or other techniques? Parallelizing seems not so effective in this example.

Reviewer 3 ·

Basic reporting

No comment

Experimental design

No comment

Validity of the findings

In this paper, the authors formulate and verify security
properties of the Hybrid Post-Quantum TLS protocol
in Maude-NPA and its parallel version. The paper starts
with a brief overview of Maude and the way narrowing works
in the system with the help of an example. The Hybrid
Post-Quantum TLS protocol is then first described and then
formalized in Maude-NPA. Towards this last goal, the authors
present the concept of strands, that is used to specify
the protocol and the capabilities of the intruder
(using the standard Dolev-Yao model) in Maude-NPA.
Correctness of the resulting Maude-NPA specification
is ensured by formally proving that it is possible to complete
the protocol handshake between two principals. The authors then
present the results of checking three security properties both
with Maude-NPA and parallel Maude-NPA. For the first of these, the
parallel Maude-NPA is able to find a counterexample of depth 12,
while Maude-NPA fails to give a definite response within the time
bound of approx. 72 days. For the second and third properties,
both Maude-NPA and its parallel version could not find a
counterexample within the time bound, thus ensuring that the attack
is unsuccessful for the depth reached within the check. The authors
also demonstrate how increasing the number of workers in the parallel
Maude-NPA impacts on performance. The paper ends with a thorough
analysis of related work.

I like the paper very much. Maude-NPA supports a declarative
style for the specification of the protocol execution, of the
intruder capabilities and of the attack pattern, which makes
matter simple and elegant. The presentation is very good and
accessible for a researcher in the field of cryptographic
protocols without previous knowledge in Maude. The experiment
is described in detail and the data are made available for
possible replication (of course, I did not have access either
to a supercomputer or 72 days to wait for the results).
I think this is a fine contribution and the submission should
be accepted after having solved the rather minor things below.

My most important criticism has to do with the provided sources.
I have taken a look at the Maude-NPA specification at
https://github.com/duongtd23/PQTLS-MaudeNPA/blob/main/HB-TLS.maude.
I think it would have been better to:
- remove the generic comments, e.g.
"Overwrite this module with the algebraic properties
of your protocol",
- add more informative comments about the code instead,
- include ATTACK-STATE(0), -(1) and -(2) at least as
comments.

Other comments for the authors:
- line 97: Maude does not specify equational and
rewriting logic, rather these are the underlying
logic of its functional and system modules,
respectively.
- line 99: "... are equipped... " ~> "... are implemented ..."
- line 445 and line 473: you point to Section 1
where you have a reference (in the bibliography) to a
webpage. I think it would be easier to give the URL
directly in both places.
- the sort of ATTACK-STATE(10) is not given, and only
informally described. Perhaps it would make the paper
more complete to present the five component constructor
of the sort in Maude.
- line 631: "the data shown in Table 2 are exactly
the same up to depths 10, 10, and 17 for the three
attack patterns 0, 1, and 2, respectively,
with Maude-NPA and Par-Maude-NPA" - does this mean that
the number of reachable states is the same for Maude-NPA
and Par-Maude-NPA? I don't find this surprising.
- line 672: I don't think you can compare theorem proving
(which of course requires introduction of lemmas) with
narrowing. It would be interesting to add a remark about
the possibilities of Maude-NPA to interact with theorem
provers, other than via CafeOBJ proof scores.

---

## Round 0.2 · Minor Revisions

When you submit your final version, please take care of the minor comments raised by Reviewer2, in particular the following:

1. The title: Maude-NPA and Par-Maude-NPA are two extensions of Maude. I'm not sure if it is a good idea to include them in the title because they don't provide much information. Instead, it would be better to reflect some essential techniques behind the tools in the title, allowing readers to quickly grasp the insights of the work and understand what makes the verification successful.

2. Please consider whether it is a good place to put Section 4.2. I recommend placing it in the preliminaries section.

3. Please clarify why the Maude code in Sect. 4.4 guarantees the correctness of the specification. Specifically, the code only guarantees that the handshake can be completed, but it doesn't necessarily ensure overall correctness.

·

Basic reporting

The authors have followed the recommendations and answered the questions posed to them.

Experimental design

The explanations about the experiments have been improved.

Validity of the findings

No comment.

Reviewer 2 ·

Basic reporting

The paper has been significantly improved compared to the previous version. My comments have been addressed and reflected in the revised version. This paper presents a compelling case study on formal modeling and verifying an important real-world hybrid post-quantum transport layer security protocol. The paper's motivation is clear, and the case study is solid. The verification results are reasonable, demonstrating the effectiveness of the proposed method and the power of the verification tools. Additionally, finding a counterexample is encouraging and important for the protocol developers to repair and improve the protocol's security.

I only have several minor comments:

1. The title: Maude-NPA and Par-Maude-NPA are two extensions of Maude. I'm not sure if it is a good idea to include them in the title because they don't provide much information. Instead, it would be better to reflect some essential techniques behind the tools in the title, allowing readers to quickly grasp the insights of the work and understand what makes the verification successful.

2. Please consider whether it is a good place to put Section 4.2. I recommend placing it in the preliminaries section.

3. Please clarify why the Maude code in Sect. 4.4 guarantees the correctness of the specification. Specifically, the code only guarantees that the handshake can be completed, but it doesn't necessarily ensure overall correctness.

Experimental design

The experimental design is reasonable. However, the experimental results can be improved.
Section 5: It would be better to have two subsections for effectiveness (5.1, 5.2, 5.3) and efficiency (5.4), respectively.

Validity of the findings

no comments

Reviewer 3 ·

Basic reporting

I am generally happy with the responses to my comments, and I believe the additions answer the issues raised by the other reviewers in a satisfactory manner. I maintain my opinion that this is an interesting contribution and it deserves to be published.

One rather minor comment for the authors:

- "Maude allows users to directly specify both order-sorted equational logic and rewriting logic (Meseguer, 2010)." Sorry to insist on this, but users don't specify the logics. I think the right formulation is "Maude allows users to directly specify both in order-sorted equational logic and rewriting logic"

Experimental design

See above

Validity of the findings

See above

---

## Round 0.3 · accepted · Accept

I have checked the revision and feel that it has addressed the minor revision comments raised by the reviewers and feel that the manuscript is ready for publication.